# Translational Regulation by hnRNP H/F Is Essential for the Proliferation and Survival of Glioblastoma

**DOI:** 10.3390/cancers14051283

**Published:** 2022-03-02

**Authors:** Morgane Le Bras, Noah Gorelick, Sylvain Pautet, Betty Tyler, Stéphane Manenti, Nicolas Skuli, Stefania Millevoi, Anne Cammas

**Affiliations:** 1Cancer Research Centre of Toulouse, INSERM UMR 1037, 31037 Toulouse, France; morga.lebras@wanadoo.fr (M.L.B.); stephane.manenti@inserm.fr (S.M.); stefania.millevoi@inserm.fr (S.M.); 2Université Toulouse III Paul Sabatier, 31330 Toulouse, France; sylvain.pautet@acimmune.com; 3Hunterian Neurosurgical Research Laboratory, Department of Neurosurgery, Johns Hopkins University School of Medicine, Baltimore, MD 21205, USA; noahgo@uw.edu (N.G.); btyler@jhmi.edu (B.T.); nicskuli@pennmedicine.upenn.edu (N.S.)

**Keywords:** glioblastoma, translation, RNA-binding protein, hnRNP H/F, RNA G-quadruplex

## Abstract

**Simple Summary:**

Developing effective treatments for glioblastoma (GBM), a highly aggressive brain tumor that is resistant to current therapies, is an urgent medical need that can be addressed by the in-depth study of basic biology for the identification of relevant targets. The aim of the present study was to investigate the molecular mechanisms underlying the deregulation of protein synthesis associated with GBM progression and resistance to treatments. Our present work demonstrates the role of the RNA-binding proteins hnRNP H/F as key players in the control of protein synthesis in GBM through different overlapping mechanisms. Furthermore, our results show that hnRNP H/F potentiate cellular processes underlying the aggressive and resistant phenotype of GBMs, thus indicating hnRNP H/F as a potential target for therapeutic intervention.

**Abstract:**

Deregulation of mRNA translation is a widespread characteristic of glioblastoma (GBM), aggressive malignant brain tumors that are resistant to conventional therapies. RNA-binding proteins (RBPs) play a critical role in translational regulation, yet the mechanisms and impact of these regulations on cancer development, progression and response to therapy remain to be fully understood. Here, we showed that hnRNP H/F RBPs are potent regulators of translation through several mechanisms that converge to modulate the expression and/or the activity of translation initiation factors. Among these, hnRNP H/F regulate the phosphorylation of eIF4E and its translational targets by controlling RNA splicing of the *A-Raf* kinase mRNA, which in turn modulates the MEK-ERK/MAPK signaling pathway. The underlying mechanism involves RNA G-quadruplex (RG4s), RNA structures whose modulation phenocopies hnRNP H/F translation regulation in GBM cells. Our results highlighted that hnRNP H/F are essential for key functional pathways regulating proliferation and survival of GBM, highlighting its targeting as a promising strategy for improving therapeutic outcomes.

## 1. Introduction

Glioblastoma (GBM) is a grade IV brain tumor, which is one of the most aggressive primary brain tumors in humans [1]. Despite the current standard of care, combining surgical resection with radiotherapy and concomitant/adjuvant chemotherapy with temozolomide (TMZ), the prognosis of patients with GBM remains poor, with a median survival of 14–15 months [2,3]. This dismal outcome is mainly due to a high rate of tumor recurrence linked to the characteristic heterogeneous population of cells that are genetically unstable, highly infiltrative and resistant to conventional treatments [4]. In this context, the still incomplete understanding of the basic biology of GBM needs to be investigated to identify new targets and develop novel therapeutic strategies to fill this major unmet clinical need in oncology.

mRNA translation is the most energetically consuming process in the cell, whose deregulation is widely recognized as a hallmark of cancer, supporting its development and progression [5,6]. Being a focal point of almost all major oncogenic signaling pathways, including Ras-MAPK and PI3K-AKT-mTOR, translation serves as a crucial node in cancer pathways. Accordingly, understanding how translation is regulated in cancer cells and identifying strategies to target it for therapeutic purposes [7,8,9,10] has become a major challenge in both fundamental and translational research. There is evidence that the expression and/or activity of translation regulators—such as eukaryotic translation initiation factors (eIFs) or RNA-binding proteins (RBPs)—are modified in cancer [11,12,13,14]. Alterations in translational control lead to a reprogramming of the genome by steering not only a global increase in protein synthesis rates supporting cellular proliferation and growth, but also specific gene expression programs that drive distinct aspects of cancer cell behaviors [5,6]. In particular, the eukaryotic translation initiation factor 4E (eIF4E) is hyperactivated via phosphorylation downstream of the MAPK-interacting serine/threonine kinases (MNKs) [15,16] and by mTOR-dependent inactivation of its negative regulators—tumor-suppressive eIF4E-binding proteins (4E-BPs) [17]. The regulation of eIF4E expression or phosphorylation drives oncogenesis by selectively increasing the translation efficiency of subsets of mRNAs encoding protumorigenic factors [5,15,16]. 

In GBM cells, accumulating data has revealed that translational regulation is hijacked to promote tumor proliferation and/or develop resistance to treatments. Notably, the expression of several eIFs, including eIF3e [18,19], 3c [20] and 5A [21], or RBPs [12,22,23], is altered and mutations in RBP-binding domains have been reported in GBM [22]. More specifically, the overexpression of eIF4E [24] and an increase in its phosphorylation [25] correlates with higher tumor grade and is associated with poorer prognosis [25]. In response to different treatments (irradiation, TMZ or arsenic trioxide), translational deregulations drive selective translation of mRNA-encoding factors involved in the DNA damage response (DDR) [26,27,28] or related to eIF4E activation [28,29,30]. Recently, new insights into the mechanisms underlying the selective translation of DNA repair factors in GBM have pointed out the crucial role of the RBPs hnRNP H1/F impacting the cellular stress response linked to the outcome of GBM [26]. 

Members of the hnRNP H/F family (including hnRNP H1, hnRNP H2, HnRNP H3, hnRNP F) are among the most deregulated RBPs in several types of cancers [12,31], including GBM [26,32], where they represent likely clinically relevant molecular targets [26]. Structurally and functionally related, hnRNP H1 (hereafter called hnRNP H) and F are mainly known for their function in regulating mRNA processing including polyadenylation and mRNA splicing through a direct interaction with G-rich RNA motifs, presenting capacities to form RNA structures called G-quadruplexes (RG4s) [33,34]. Their role as translational regulators has been studied to a lesser extent and the investigation of the molecular mechanism underlying their function in this post-transcriptional step is still in its infancy [35,36]. Our recent work revealed that hnRNP H/F, together with the RNA helicase DHX36, holds the RG4s unfolded, resulting in translational regulation of a specific group of DDR mRNAs and contributing to resistance to treatments in GBM [26]. These results, together with studies showing that the splicing activity of hnRNP H affects Ras-MAPK pathways in cancer [37] and the expression of translation factors [38], prompted us to investigate whether—and if so, how—hnRNP H/F affects mRNA translation by regulating the expression and/or the activity of translation factors, and to explore the consequences of these regulations in GBM. Here, we propose a model in which RG4- and hnRNP H/F-dependent splicing regulation of *A-Raf* controls the Raf-MEK-ERK/MAPK pathway, resulting in eIF4E phosphorylation. This and other mechanisms affecting the expression of key translation factors allow hnRNP H/F to control global and selective translation to drive key functional pathways involved in GBM progression and response to treatments.

## 2. Materials and Methods

### 2.1. Cell Culture and Treatment

The glioblastoma cells LN18, U251-MG and U87 were obtained from ATCC (CRL-2610), ECACC (#09063001) and SIGMA (#89081402-1VL), respectively. They were maintained in DMEM media containing 4.5 g/L glucose and supplemented with 10% FBS, 2 mM L-glutamine, 100 U/mL penicillin and 100 µg/mL streptomycin. Mycoplasma contamination was frequently assessed by PCR. Where indicated, cells were treated at 37 °C with cPDS (Sigma-Aldrich, St. Louis, MO, USA, SML1176), puromycin (Sigma P8833), cycloheximide (Sigma-Aldrich C7698), etoposide (Sigma-Aldrich E1383), with the indicated concentration and for the indicated time.

### 2.2. Cell Transfection

siRNAs were transfected using the Lipofectamine^TM^ RNAiMAX (Thermo Fisher Scientific, Waltham, MA, USA) according to the manufacturer’s instructions. In brief, cells were reverse-transfected with 2.5 nM of siRNAs which were synthesized by SIGMA. Cells were subsequently incubated at 37 °C for 72 h before harvesting and analysis. LN18 cells with stable silencing were generated by transduction of lentivectors (plasmids MISSIONH pLKO.1-puro, Sigma-Aldrich), expressing shRNA control (SHC002), or shRNAs against hnRNP H and hnRNP F. All siRNA and shRNA sequences are available in Appendix A.

### 2.3. Western Blotting

For immunoblotting analysis, proteins were resolved on 12 or 7% denaturing polyacrylamide gels and were transferred to nitrocellulose membranes. The blots were blocked for 30 min with TBS-T-5% milk and then probed overnight with primary antibodies against hnRNP H/F (1:1000, Abcam, Cambridge, UK, Ab10689), A-Raf full length (1:1000, Cell signaling, Danvers, MA, USA, 4432), A-Raf short (polyclonal anti-human antibody kindly provided by J. Rauch and generated as in [37]) p-ERK 1/2 (1:2000, Cell signaling 9106S), ERK 1/2 (1:1000, Cell signaling 9102S), p-eIF4E (1:2000, Abcam ab76256), eIF4E (1/1000, Santa Cruz Biotechnology, Dallas, TX, USA, sc-9742S), hnRNP A2/B1 (1/1000, Santa Cruz sc-53531), hnRNP A1 (1/1000, Santa Cruz sc-32301), hnRNP I (HB-94, cloneBB7.7; ATCC), MMP-9 (1:1000, Abcam Ab13458), SNAIL (1:1000, Cell signaling 3879S), eIF4G (1/1000, Santa Cruz sc-4373), eIF4B (1:1000, Cell signaling 3592), eIF4H (1:1000, Cell signaling 3469), 4E-BP (1:1000, Cell signaling 9644S), GAPDH (1:1000, Santa Cruz sc-32233), Puromycin (1:1000, Millipore, Burlington, MA, USA, MABE343), PARP (1:1000, Cell signaling 9542), and Caspase-3 (1:1000, Cell signaling 8G10), as well as secondary anti-rabbit (1:5000, Ozyme, Saint-Cyr-l’École, FR, 7074S) and anti-mouse (1:5000, Ozyme 7076S) IgGs. The blots were developed using the ECL system (Amersham Pharmacia Biotech, Amersham, UK) according to the manufacturer’s directions and images quantified using FIJI software. All original western blot can be found in Appendix A.

### 2.4. RT-qPCR and RT-PCR

A measure of 1 µg of total RNA was quantified using the Clariostar BMG (v.5.21 R4) combined with the MARS Clariostar Analysis Software (v.3.20 R2) and were reverse transcribed using the RevertAidH Minus First (Thermo fisher, Waltham, MA, USA) following the manufacturer’s rules. qPCR analysis of cDNA (12.5 ng) was performed with the SybrGreen (KAPA KK4605) using the StepOne Applied Biosystems, Waltham, MA, USA (v2.2.2). Expression of indicated mRNAs was normalized to the *GAPDH* reference mRNA, and relative levels of expression were quantified using the 2^ΔΔCT^ method, where CT is cycle number at which the amount of amplified target reaches a fixed threshold. For RT-PCR, PCR conditions were 25 cycles of denaturing at 95 °C for 30 s, annealing for 20 s, and extension at 70 °C for 20 s. All primer sequences are available in Appendix A.

### 2.5. Immunoprecipitation

Cell extracts were obtained as described in [26], except that the pellet fraction was resuspended in 500 μL lysis buffer A (10 mM Tris pH 8.0, 140 mM NaCl, 1.5 mM MgCl_2_, 0.5% NP40, 1 mM DTT) supplemented with 0.1% SDS. After centrifugation at 13,000 × *g* (4 °C) for 5 min, supernatant (nuclear soluble fraction) was transferred into a fresh tube. Cell extracts were treated with Benzonase (Millipore E1014) and DNase I (Thermo Scientific, Waltham, MA, USA, EN0521) for 1 h at room temperature, and then precleared on protein-Sepharose beads for 1 h at 4 °C. BG4 (0.5 µg, expressed from the pSANG10-3F-BG4 plasmid (Addgene, Watertown, MA, USA, #55756), kindly provided by S. Balasubramanian) or hnRNP H/F (10 µg, Abcam Ab10689) antibodies were incubated with 20 µL of slurry beads (washed and equilibrated in cell lysis buffer) for 1 h at 4 °C. A measure of 1 mg of cell extracts was added on beads and incubated on a wheel overnight at 4 °C. After five washes of the beads with cell lysis buffer, the immunoprecipitated proteins and RNAs were eluted in the NT2 buffer (50 mM Tris pH 7.4, 1 mM MgCl_2_, 0.05% NP-40). 

### 2.6. Preparation of RNA–Protein Complexes and Analysis by RT-qPCR

Eluted samples from the immunoprecipitation were treated with proteinase K (Euromedex). RNAs were extracted with Phenol/Chloroform and resuspended in 10 µL of water. A measure of 4 µL was reverse transcribed using the RevertAidH Minus First (Thermo fisher) following the manufacturer’s indication. A 1/5 dilution of cDNA was used to analyze mRNA levels by qPCR with the SybrGreen ( Sigma-Aldrich, St. Louis, MO, USA, KK4605). The quantities of mRNA contained in these mRNP complexes were then normalized to the quantity of *HPRT* (reference mRNA) and compared with the RNA levels contained in the IgG control and input sample.

### 2.7. SUnSET

Cells were incubated with 10 µg/mL of puromycin (Sigma P8833) for 10 min at 37 °C. After 2 washes in ice-cold PBS, cells were scraped on ice in PBS, pelleted by centrifugation at 200 × *g* for 5 min and lysed in the lysis buffer (0 mM HEPES pH 7.0, 150 mM NaCl, 10% Glycerol, 1% Triton, 10 mM Na4P2O7, 100 mM NaF, 1 mM EDTA with 1.5 mM MgCl_2_ and 10 µL/mL Protease Cocktail Inhibitor (Sigma, P8340)). Puromycin incorporation was detected using western Blot analysis.

### 2.8. Polysome Profiling

Polysome profiling was performed as previously described [26]. Briefly, cells that should not exceed 80% of confluency, were incubated for 15 min at 37 °C with 0.1 mg/mL cycloheximide (CHX) and lysed in 450 μL of hypotonic lysis buffer [5 mM Tris pH 7.5, 1.5 mM MgCl_2_, 1.5 mM KCl, 0.1 mg/mL CHX 20 U/mL RNaseOUT (Thermo Fisher Scientific, Waltham, MA, USA, 10777019), 0.5% of Triton X-100, 0.5% sodium deoxycholate and 10 µL/mL of Protease Cocktail Inhibitor (Sigma, P8340)]. A volume of lysate corresponding to 20 OD_260nm_ was loaded on a continuous 5–50% sucrose gradient and subjected to ultracentrifugation at 222,228 × *g* in a SW41-Ti rotor for 2 h at 4 °C. Sucrose gradient was fractionated with an ISCO density gradient fractionation system (Foxy Jr fraction collector coupled to UA-6UV detector, Lincoln, NE, USA) and the absorbance at 254 nm was registered. Fractions were flash-frozen immediately after fractionation and stored at −80 °C.

### 2.9. Proliferation Assay

GBM cells transfected with siRNAs or stably expressing shRNAs were harvested and counted with a coulter counter (Beckman Coulter, Brea, CA, USA) every day for 7 days. 

### 2.10. Colony Formation Assay

After 72 h of siRNA transfection, cells were recovered and seeded in 6-well plates at different concentrations (500, 750, 1000 cells/well for siRNA control and 1500, 2500, 5000 for siRNAs against hnRNP H and hnRNP F). Cells were incubated at 37 °C until colonies were visible with the naked eye (approximatively 10 days). Colonies were then fixed with 10% formalin for 10 min and incubated with 10% crystal violet oxalate (RAL Diagnostics, Martillac, France) for 10 min at room temperature. Wells were washed with water until colonies were visible for counting. The plating efficiency was defined as the percentage of number of colonies formed over number of cells plated.

### 2.11. Xenograft Tumors in Nude Mice

Animal experiments were performed in the neurosurgical department of Johns Hopkins University in Baltimore. Briefly, LN18 cells (1.5 × 10^6^) were mixed with PBS/Matrigel (ratio 1:1) and subcutaneously injected in each flank of nude mice (*n* = 4 mice per condition). Tumor volumes were measured every day for 40 days. All procedures were performed in accordance with the guidelines set forth by the Johns Hopkins University Animal Care and Use Committee.

### 2.12. Flow Cytometry

After 72 h of siRNA transfection, LN18 cells were collected, washed in PBS and prepared for flow cytometry. For cell death, cells were incubated with 5 µL of AnnexinV-PE and 1 µL of SYTOXGreen solution at 1 µM for 10 min at room temperature and protected from light (Annexin V-PE Apoptosis Kit Plus, Biovision, Waltham, MA, USA). For cell cycle analysis, cells were fixed with cold ethanol 70% and permeabilized with a solution of PBS/0.025% triton for 15 min on ice. Then, cells were labelled with propidium iodide (20 µg/mL) for 15 min at 37 °C with RNAse A at 10 µg/mL. Cells were kept on ice until the run on a flow cytometer (MACSQuant^®^VYB, Miltenyi Biotec, San Diego, CA, USA). Subsequent analyses were performed using FlowJo software (FlowJo LLC, Ashland, OR, USA) 

### 2.13. Migration/In Vitro Wound Closure Assay

Cells were plated in 6-well plates at a density allowing to reach confluency after overnight incubation at 37 °C. The wound was realized by scraping the cell layers with a plastic pipette tip. After 3 washes with serum-free medium, the remaining cells were incubated for 20 h at 37 °C allowing the migration into the cleared space. Phase contrast images of identical positions in each wound were taken to allow the measurement of wound closure. 

## 3. Results

### 3.1. hnRNP H/F Impact on A-Raf Splicing and eIF4E Phosporylation 

Previous results showed that a reduction in hnRNP H expression switches the splicing of *A-Raf* mRNA to produce A-Raf short-truncated form of the kinase which acts as a dominant–negative Ras antagonist that negatively regulates the Raf-MEK-ERK/MAPK pathway [37]. However, whether and how hnRNP H (and possibly hnRNP F) impacts the phosphorylation of eIF4E, which is downstream of this signalling pathway, has not been demonstrated. To investigate this link, we started to define whether hnRNP H/F controlled *A-Raf* splicing in GBMs. To this end, we transfected LN18 GBM cells with hnRNP H and/or hnRNP F-specific or control siRNAs, followed by quantification of *A-Raf* mRNA isoforms using specific primers (Appendix A). We found that hnRNP H, hnRNP F or hnRNP H/F (i.e., both hnRNP H and hnRNP F) silencing induced a 4.9-, 1.7- or 18-fold increase in *A-Raf short* mRNA levels, respectively (Figure 1A and Appendix A), while *A-Raf full-length* mRNA levels were reduced (Appendix A). Moreover, we observed an increase in A-Raf short protein correlated with a decrease in A-Raf full length protein following hnRNP H, hnRNP F or hnRNP H/F depletion (Appendix A), suggesting that the hnRNP H/F-mediated regulation of *A-Raf* mRNA splicing affects A-Raf expression both at the RNA and protein level. Then, we tested whether this effect was linked to hnRNP H/F binding to *A-Raf* mRNA by performing in vitro RNA immunoprecipitation (RIP) assays using nuclear extracts (Appendix A) from GBM cells. We observed that hnRNP H/F antibody significantly immunoprecipitated *A-Raf* mRNA as compared with control IgG, with an extent similar to the positive control *USP1* [26] (Figure 1B and Appendix A), suggesting that *A-Raf* splicing is mediated by the formation of ribonucleoprotein complexes involving hnRNP H/F.

In agreement with previous findings reporting that A-Raf short protein negatively regulates the Ras-MAPK(Raf-MEK-ERK) pathway [37], we found that hnRNP H and/or hnRNP F silencing significantly decreased the phosphorylation of ERK 1/2 compared with the total ERK 1/2 amount in LN18 GBM cells (Figure 1C,D). Based on the observation that the Ras-MAPK pathway activation impacts on eIF4E activity [39], we then investigated whether silencing of hnRNP H and/or hnRNP F would affect eIF4E phosphorylation on serine 209. Our results showed that hnRNP H/F depletion significantly inhibited the phosphorylation of eIF4E on serine 209, while leaving the expression of total eIF4E unaffected in LN18 GBM cells (Figure 1C,D). Similar results were observed in U87 and U251 (Appendix A) GBM cells, indicating that the effects of hnRNP H/F on this signaling pathway were not cell-type-specific effects. Taken together, these results suggest that hnRNP H/F-mediated control of *A-Raf* splicing in GBM cells increases eIF4E phosphorylation, possibly impacting on its function in translational regulation. 

### 3.2. Role of RG4 Stabilization on A-Raf Splicing and eIF4E Phosphorylation 

Since hnRNP H/F play a role in post-transcriptional regulation by binding and modulating RG4 formation [26,33,34], we investigated whether hnRNP H/F impact on *A-Raf* splicing and eIF4E phosphorylation (Figure 1) might involve an RG4-dependent mechanism. Indeed, we previously showed that hnRNP H/F depletion increased RG4 structuration and functionally mirrored the effect of RG4 stabilization by small-molecule ligands specific to RNA G-quadruplexes, namely cPDS (carboxypyridostatin) [26,40]. To test this possibility, we first assessed the ability of cPDS to increase RG4 formation in *A-Raf* transcripts by RIP assays with LN18 cytoplasmic extracts and the BG4 antibody, known to recognize RG4s [40]. Our data showed an increase in *A-Raf* mRNA interaction with the BG4 antibody following treatment with cPDS to an extent similar to the RG4-containing *USP1* mRNA [26] (Figure 2A). This observation led us to conclude that *A-Raf* mRNA is prone to form RG4s in cellulo. Then, we analyzed the impact of RG4 stabilization on *A-Raf* mRNA expression, Ras-MAPK pathway and eIF4E phosphorylation. We observed that cPDS induced a significant increase in *A-Raf short* mRNA levels (Figure 2B) and protein levels (Figure 2C,D) accompanied by a reduction in ERK 1/2 and eIF4E phosphorylation, whereas the total amount of protein was not affected (Figure 2C,D). Taken together, these results revealed that the splicing of *A-Raf* mRNA linked to Ras-MAPK pathway and impacting eIF4E phosphorylation depends on a tight control of the RG4-folding equilibrium.

### 3.3. Role of hnRNP H/F in Translational Control in GBM

To further explore the functional consequences of hnRNP H/F-mediated control of eIF4E phosphorylation, we analyzed the effect of downregulating hnRNP H and hnRNP F on the expression of proteins known to be translationally regulated by eIF4E phosphorylation [15,16]. Our results showed that hnRNP H and hnRNP F depletion induced a significant decrease in the protein amount of MMP-9 and SNAIL, two well-known translation targets of phosphorylated eIF4E [15,16] (Figure 3A,B), suggesting that hnRNP H/F effect on translation is mediated by the control of eIF4E phosphorylation.

To bring further insights into the translational role of hnRNP H/F in GBM, we quantified global protein synthesis rates in LN18 GBM cells pulse, labeled with puromycin (i.e., SUnSET assay) and transfected with control-, hnRNP H- and/or hnRNP F-specific siRNAs. Our data revealed that hnRNP H or hnRNP F depletion induced a significant 40% reduction in global translation rate comparable to the effect of downregulating the translational regulators hnRNP A1 and hnRNP I [41] (Figure 3C,D). Strikingly, the downregulation of both proteins in LN18 significantly and drastically reduced global translation by 80% (Figure 3C,D). Of note, a similar effect of hnRNP H and/or hnRNP F on global protein synthesis rates was observed using different siRNAs in LN18 cells (Appendix A), ruling out potential siRNA off-target effect. To confirm and further validate the SUnSET analysis, we assessed translation efficiency with polysome profiling experiment. We found that polysome profile is drastically affected by hnRNP H and hnRNP F depletion in LN18 cells (Appendix A). Altogether, these results indicate that cells deficient in hnRNP H/F are globally defective in protein synthesis, suggesting that, in addition to controlling eIF4E phosphorylation, the impact of hnRNP H/F on translation involves additional mechanisms. 

Based on the effect of hnRNP H/F on global translation (Figure 3C,D) and considering previous results showing that hnRNP H/F associate with mRNAs linked to the “cytoplasmic translation” gene ontology term [26], regulate the splicing of translation factors [38,42] and interact with mTOR kinase [43], we tested whether hnRNP H/F might affect global translation by regulating the expression of other eIFs or the mTOR activation. Together with eIF4E, the scaffold protein eIF4G, and the DEAD-box helicase eIF4A, form a complex called eIF4F that recognizes the cap through the binding of eIF4E and unwinds the secondary structures during the scanning of the 5′UTR via the helicase activity of eIF4A stimulated by the auxiliary factors eIF4B and eIF4H, until the recognition of the start codon [44]. Our data showed that hnRNP H/F depletion induced a switch or a decrease in eIF4H isoform expression in LN18 cells, while eIF4G and eIF4B, two other reported hnRNP H/F targets [45], were unaffected (Appendix A). Moreover, we found that hnRNP H depletion induced the hypophosphorylation of the mTOR target, 4E-BP, in LN18 cells (Appendix A). Altogether, these results underscored the role played by hnRNP H/F in the control of global translation rates in GBM cells and suggest that the mechanism underlying this function involves the modification of the expression or the activity of eIFs factors.

### 3.4. Functional Impact of hnRNP H/F on GBM Cell Migration and Proliferation 

Based on the regulatory role of hnRNP H/F on metastasis-related mRNA expression (Figure 3A,B), and in agreement with previous findings showing that phosphorylation of eIF4E promotes metastasis via translational control [16], we decided to test whether hnRNP H/F might influence cell migration in a scratch wound assay. We found that the depletion of hnRNP H and/or hnRNP F significantly impaired LN18 (Figure 4A,B) or U251 (Figure 4C,D) cells’ ability to close the wound 20 h after the scratch. These data suggest that the translational regulation of metastasis-related proteins induced by hnRNP H/F-mediated control of eIF4E phosphorylation might affect cell migration in GBM cells. 

Since alterations in the mechanisms regulating translation affect cancer cell proliferation [6], we then sought to investigate whether hnRNP H/F regulate GBM cell proliferation. To address this question, we transfected GBM cells with hnRNP H- and/or hnRNP F-specific or control siRNAs and followed cell growth by counting cells daily for 8 days. Three–four days after transfection, we observed a significant decrease in cell number after hnRNP H and/or hnRNP F silencing in LN18 (Figure 5A), U87 (Appendix A) and U251 (Appendix A) cells. Of note is that the effect of hnRNP H depletion is comparable to the one of hnRNP H/F depletion and appeared to be stronger than the effect of hnRNP F silencing on GBM cell proliferation (Figure 5A). Similar results were obtained by silencing hnRNP H and hnRNP F using specific shRNA (Appendix A), excluding the potential siRNA off-target effects. In good agreement with this, ectopic expression of hnRNP H and hnRNP F in U87 cells significantly increased cell proliferation (Figure 5B). To further validate the functional role of hnRNP H/F in GBM cell proliferation, we assessed the number of foci formed by LN18 cells transfected with hnRNP H- and/or hnRNP F-specific or control siRNAs, in a colony formation assay. Our results showed that LN18 ability to form colony was significantly impaired showing a 95% and a 30% reduction in plating efficiency after hnRNP H and hnRNP F depletion, respectively (Figure 5C). In agreement with the proliferation assay results (Figure 5A), we observed a more deleterious effect of hnRNP H depletion as compared with hnRNP F downregulation (Figure 5C). Taken together, these results indicate that hnRNP H/F are essential for cell growth and proliferation in GBM cells.

To provide further insights into the effect of hnRNP H/F on GBM cell growth, we first assessed cell cycle changes in LN18 cells treated with either hnRNP H- and/or hnRNP F-specific or control siRNAs and followed propidium iodide (PI) staining using flow cytometry. Our data revealed that cells depleted for hnRNP H were accumulated in G2 phase (22.2% ± 2.9) compared with control cells (16.5% ± 0.7), while hnRNP F depleted cells accumulated in G1 phase (71.7% ± 3.9) compared with control cells (58.3% ± 0.7) (Appendix A). Of note, the depletion of both hnRNP H and hnRNP F showed a tendency toward an intermediate effect with an increase in the number of cells in G1 (65.8% ± 3.2) and G2 (24.9% ± 2.8) phases (Appendix A). Overall, these results suggest that the decrease in proliferation induced by hnRNP H/F silencing involves a cell cycle arrest in G1 and G2 phases of the cell cycle. Since cell death and apoptosis can occur in response to cell cycle arrest [46], we then investigated the impact of hnRNP H/F on apoptosis in GBM cells measured with flow cytometry analysis of Annexin-V/SYTOXGreen staining. We observed that the apoptotic cell death rate was significantly increased in hnRNP H- (2.5-fold) and in hnRNP F-depleted (5.5-fold) cells (Appendix A). Remarkably, the depletion of both hnRNP H and hnRNP F induced an even greater increase in the number of apoptotic GBM cells (8-fold). Notably, the increase in the SYTOXGreen staining alone indicated that the number of necrotic cells increased upon hnRNP H and/or hnRNP F silencing in LN18 cells (Appendix A). In addition, the observation of an increase in PARP and caspase3 (apoptotic proteins) cleavage, comparable to the one induced by etoposide treatment in LN18 cells, confirmed the effect of hnRNP H/F downregulation on apoptosis induction and the synergistic effect of both proteins on this cellular process (Appendix A). Overall, these findings strongly suggest that hnRNP H/F depletion inhibits GBM cell proliferation through induction of cell cycle arrest, which would precede cell death processes.

To further confirm our in vitro data and explore whether the effect of hnRNP H/F on cell growth and proliferation finds an echo in the tumoral development in vivo, we developed a xenograft tumor model by subcutaneously injecting in each flank of a nude mouse, LN18 cells in which hnRNP H/F depletion was induced by shRNA. We found that downregulating hnRNP H and hnRNP F expression significantly and drastically decreased the volume of the tumors to such an extent that only few tiny tumors were recovered in the shRNA H condition (Figure 5D). Altogether, these results supported a critical role of hnRNP H/F in the tumoral development of GBM cells in vivo. 

## 4. Discussion

RNA-binding proteins fine-tune gene expression by regulating sets of targets within specific post-transcriptional layers, in a coordinated manner, revealing their importance as a regulatory nexus to control cancer-related pathways [12,13,14]. Being deregulated in several type of cancers [12,31], and acting in a tightly controlled interplay with other RBPs [26,45] to regulate the cancer-related mRNA post-transcriptional network [26,34,42,47], hnRNP H/F are a paradigm of the cancer-related RBPs. This work revealed additional facets of the complex post-transcriptional regulatory network regulated by hnRNP H/F, contributing to multiple pathological aspects of the GBM phenotype. 

By showing that hnRNP H/F switch the splicing of *A-Raf* mRNA in GBM (Figure 1), our results not only extend this splicing event to other cancer cells than head and neck and colon cancer cells [37], but also add *A-Raf* to the list of mRNAs spliced by hnRNP H in GBM (including the adaptor protein IG20/MADD and the RON receptor tyrosine kinase [32]) and assign a novel function for hnRNP F as a splicing regulator in GBM, which has not been reported so far. While the mechanism underlying hnRNP H function on *A-Raf* splicing was not previously demonstrated [37], our observations—that (i) *A-Raf* mRNA formed a ribonucleoprotein complex with hnRNP H/F (Figure 1), (ii) *A-Raf* mRNA contained an RG4 (Figure 2) and (iii) *A-Raf short* mRNA isoform was expressed upon RG4 stabilization—support a speculative model in which hnRNP H/F regulate the switch in *A-Raf* splicing through RNA–protein interactions involving RG4 structures. The splicing of *A-Raf* mRNA was also reported to be regulated by the splicing factor hnRNP A2 in order to activate the Ras-MAPK pathway [48]. The observation that *hnRNP A2/B1* mRNA is targeted by hnRNP H/F in published in cellulo RNA–protein interactions [45,47] suggests the intriguing possibility of an additional mechanism underlying the effect of hnRNP H/F on *A-Raf* splicing involving a modulation of hnRNP A2 expression. Our results (Appendix A)—showing that (i) *hnRNP A2/B1* mRNA interacted with hnRNP H/F (Appendix A), (ii) *hnRNP A2/B1* contained G-rich sequences with the ability to fold into RG4 (Appendix A), (iii) hnRNP A2 protein levels were affected by ligand-induced RG4 stabilization (Appendix A)—suggest that the RG4-dependent regulation of hnRNP A2 expression, possibly involving hnRNP H/F binding, would impact on *A-Raf* splicing and the Ras-MAPK pathway. 

While the role of hnRNP H/F in regulating alternative splicing is well established either at a genome wide scale [34,42] or at a specific transcript level [32,47,49], their function as translational regulators have been more recently explored and less intensively investigated, so far [26,35]. The proposed model predicts that hnRNP H/F regulate the translation of specific mRNAs [35,36] or specific groups of functionally related mRNAs (regulons) [26] to steer a specific gene expression program. Our work supports this notion by showing that the hnRNP H/F-induced regulation of eIF4E phosphorylation correlated with the control of the expression of metastasis-related proteins (Figure 3). However, our results also underscore a more global effect of hnRNP H/F in protein synthesis (Figure 3). These data raise the possibility that the role of hnRNP H/F on translation involves multiple overlapping mechanisms and notably, our results revealed their implication in controlling eIF4H isoform expression (Appendix A) and 4E-BPs phosphorylation (Appendix A). 

Our results showing that hnRNP H/F-mediated splicing regulation affects translation reinforce the concept of a tight interplay between these two post-transcriptional steps resulting in the orchestrated regulation of gene expression [50,51,52]. Some RBPs have been proposed to be critical players in coordinating this link since they regulate the translation of their splicing targets [53,54]. In contrast to this mechanism, our work revealed two other layers in the interplay between splicing and translation. The first one is an indirect coupling illustrated by the hnRNP H/F-mediated splicing of *A-Raf* which impacts on translation regulatory pathways (Figure 1 and Figure 2). This “ripple effect” on different post-transcriptional steps contributes to amplify the phenotypic consequences of hnRNP H/F depletion in GBM. Second, the demonstration of a regulatory function for hnRNP H/F in the splicing of a translation initiation factor (Appendix A) not only reinforces the link between splicing and translation, but also supports the regulator-of-regulators concept [52,55]. In addition to the interplay between post-transcriptional events, the interplay between RBPs adds another layer of complexity to the post-transcriptional regulatory network [55]. Our work emphasizes this complexity through the study of the structurally and functionally related hnRNP H and hnRNP F proteins. Indeed—and in agreement with previous findings [26,33]—some of our results suggest that hnRNP H and hnRNP F behave synergistically (cooperate) or redundantly (one replace the other) in the regulation of *A-Raf* splicing (Figure 1), the Ras-MAPK pathway (Figure 1), translation (Figure 3) or apoptosis (Appendix A). Furthermore, the observation that depleting hnRNP H slightly increases the level of hnRNP F and vice versa (Figure 1C, Figure 3A,C, Figure 5A,B and Appendix A) suggests the existence of compensatory mechanisms between the two proteins that together with their functional redundancy would account for the large effect of simultaneous depletion of both proteins compared with individual protein silencing. Conversely, other findings [43] and our data showing that hnRNP H has a stronger or a distinct effect than hnRNP F on eIF4H splicing (Appendix A), 4E-BP phosphorylation (Appendix A), GBM cell migration (Figure 4), cell cycle (Appendix A) or proliferation (Figure 5) suggest the possibility of differential effects on specific post-transcriptional events involving specific mRNA targets and RBPs which need further investigation to be characterized. 

Our work, in accordance with other findings [32], demonstrates that hnRNP H/F are involved in oncogenic properties of GBM cells, from migration (Figure 4) to uncontrolled proliferation (Figure 5 and Appendix A), linked to escape from apoptosis (Appendix A) or cell cycle deregulation (Appendix A). Moreover, we previously revealed that hnRNP H/F-mediated control of translation in GBM drives response to treatments by regulating the expression of stress response genes [26]. The study showing that eIF4E phosphorylation favors cancer cell resistance to TMZ [30], together with our results (Figure 1), support interesting future investigations to determine whether the effect of hnRNP H/F on phosphorylated eIF4E constitutes one of the mechanisms underlying the impact of hnRNP H/F on GBM resistance to treatment. Additionally, the alternatively spliced eIF4H exon 5 is predicted to form a substrate generating the miR-590 [56] which is involved in the radioresistance in GBM cells by targeting the tumor suppressor LRIG1 [57]. Our results showing that hnRNP H depletion increased the expression of the exon5-containing eIF4H long isoform (Appendix A) raises questions of whether and how hnRNP H overexpression in GBM induces exon 5 splicing of eIF4H, generates miR-590 expression and contributes to GBM cell radioresistance. Finally, the observations that (1) eIF4E expression correlated with hnRNP H1, hnRNP H2, hnRNP F and A-Raf (r = 0.474, r = 0.321, r = 0.409 and r = 0.457 in the Ivy GBM database (http://glioblastoma.alleninstitute.org/ accessed on 17 May 2021), respectively), and (2) *A-Raf full length* mRNA is overexpressed in GBM primary and recurrent solid tumors (mining of the TCGA splicing variants database), support a potential clinical importance of the hnRNP H/F–A-Raf–eIF4E axis in GBM, thus making hnRNP H/F a potential target for therapeutic intervention.

## 5. Conclusions

Taken together, our data reinforce the notion that hnRNP H/F are an essential regulatory hub in GBM networks that induce a translational reprogramming affecting protein synthesis, either globally or specifically, and impact on GBM progression (Figure 6). Our work revealed the layers of complexity in the tangled mechanisms regulated by hnRNP H/F involving splicing events impacting either translation regulatory pathway or translation factor expression/activity. Finally, through a strong impact on different aspects of GBM behavior, the depletion of hnRNP H/F might be of special interest to combat GBM progression and resistance to treatments.

## Figures and Tables

**Figure 1 cancers-14-01283-f001:**
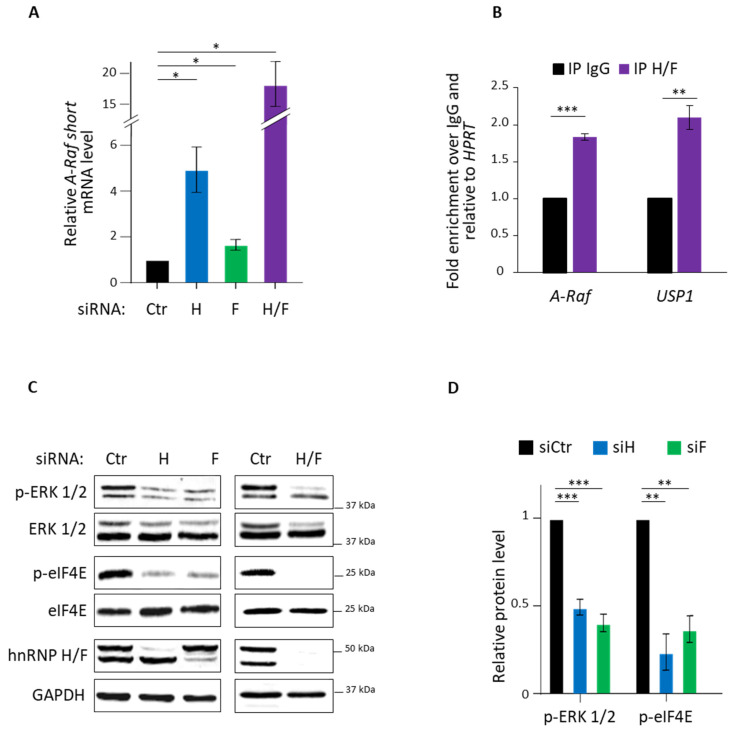
hnRNP H/F regulate *A-Raf* splicing and control eIF4E phosphorylation in GBM. (**A**) Quantitative RT-qPCR using specific primers for *A-Raf* short isoform on total cDNA generated from LN18 cells transfected for 72 h with 2.5 nM of siRNA control (siCtr), hnRNP H and/or hnRNP F (siH, siF, siH/F). *A-Raf short* mRNA levels were normalized to *GAPDH* mRNA level and to the siCtr condition. Data are plotted as mean values ± SEM of *n* = 3 independent experiments. (**B**) Immunoprecipitation of RNA–protein complexes (RIP) performed on U87 nuclear cell extracts using control IgG (IP IgG) or hnRNP H/F (IP H/F) antibody, followed by RT-qPCR analysis of *A-Raf*, *USP1* and *HPRT* mRNAs. Data are plotted as mean values ± SEM of *n* = 3 independent experiments. (**C**) Western blot analysis of phospho-ERK 1/2 (p-ERK 1/2), ERK 1/2, phospho-eIF4E (p-eIF4E) and eIF4E expression in LN18 cells treated with 2.5 nM of siRNA control (siCtr), siRNAs against hnRNP H and/or hnRNP F (siH, siF or siH/F) for 72 h. Shown is a representative result from *n* = 3 independent experiments. (**D**) p-ERK 1/2 and p-eIF4E protein levels in (**C**) were normalized to total ERK 1/2 and eIF4E protein levels, respectively, and plotted relatively to the siCtr condition. Data are presented as mean values ± SEM of *n* = 3 independent experiments for siH and siF conditions. For panels (**A**,**B**,**D**): blue, green and purple bars are respectively linked to hnRNP H, hnRNP F and hnRNP H/F; * *p* < 0.05, ** *p* < 0.005 and *** *p* < 0.0005 (two-sided paired *t*-test).

**Figure 2 cancers-14-01283-f002:**
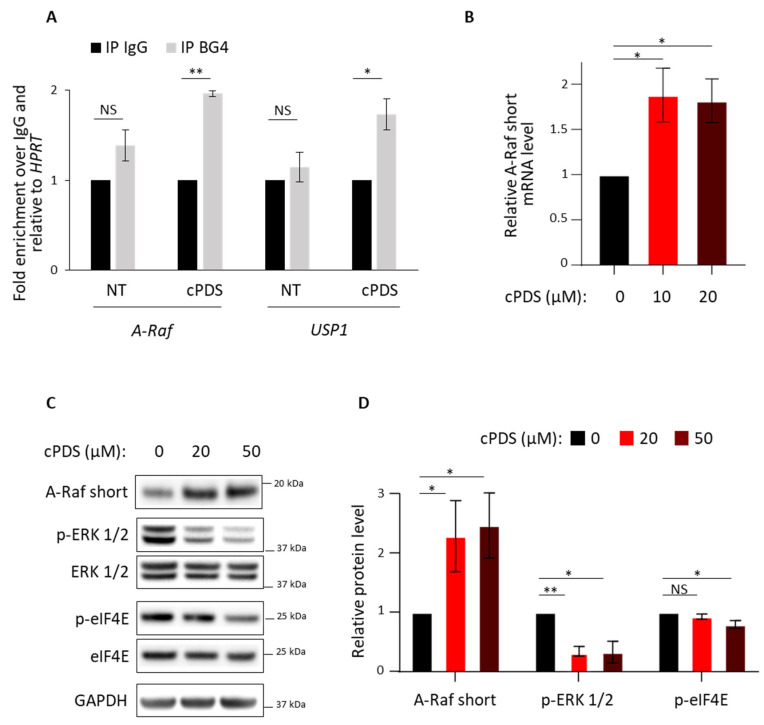
RG4 stabilization affects *A-Raf* splicing and eIF4E phosphorylation. (**A**) Immunoprecipitation (IP) of RNA–protein complexes (RIP) using control IgG (IP IgG) or BG4 (IP BG4) antibody, on cytoplasmic extracts from U87 cells untreated (NT) or treated with 20 µM carboxypyridostatin (cPDS) for 2 h and followed by RT–qPCR analysis of *A-Raf*, *USP1* and *HPRT* mRNAs. Data are plotted as mean values ± SEM of *n* = 3 independent experiments. (**B**) Quantitative RT-qPCR using specific primers for *A-Raf* short isoform on total RNA extracted from LN18 cells treated with dose scale of carboxypyridostatin (cPDS) for 4 h. *A-Raf* short mRNA levels were normalized against *GAPDH* and data were plotted relatively to the non-treated condition (0 µM cPDS). Data are presented as mean values ± SEM of *n* = 3 independent experiments. (**C**) Western blot analysis of A-Raf short, phospho-ERK 1/2 (p-ERK 1/2), ERK 1/2, phospho-eIF4E (p-eIF4E) and eIF4E expression in LN18 cells treated with dose scale of carboxypyridostatin (cPDS) for 48 h. Shown is a representative result from *n* = 3 independent experiments. (**D**) A-Raf short, p-ERK 1/2 and p-eIF4E protein levels in (**C**) were normalized to total GAPDH, ERK 1/2 and eIF4E protein levels, respectively, and plotted relatively to the non-treated condition (0 µM cPDS). Data are presented as mean values ± SEM of *n* = 3 independent experiments. For panels (**A**,**B**,**D**): red and dark red bars represent increasing concentrations of cPDS; * *p* < 0.05; ** *p* < 0.005; NS—non-significant (two-sided paired *t*-test).

**Figure 3 cancers-14-01283-f003:**
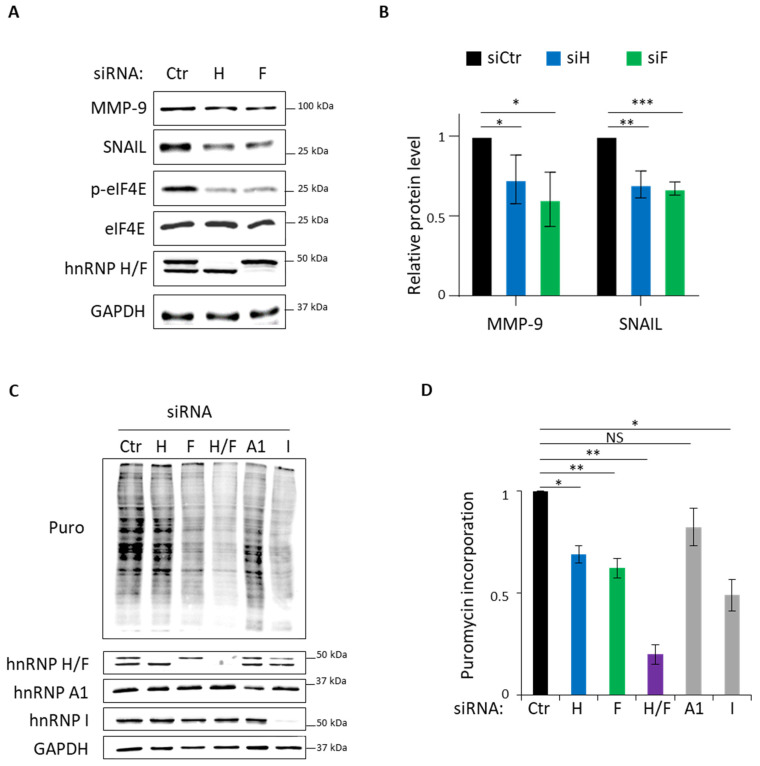
hnRNP H/F depletion inhibits translation initiation in GBM cells. (**A**) Western blot analysis of MMP-9, SNAIL, phospho-eIF4E (p-eIF4E) and eIF4E expression in LN18 cells treated with 2.5 nM of siRNA control (siCtr), siRNAs against hnRNP H and hnRNP F (siH and siF) for 72 h. Shown is a representative result from *n* = 3 independent experiments. (**B**) MMP-9 and SNAIL protein levels in (**A**) were normalized to GAPDH protein levels and plotted relatively to the siCtr condition. (**C**) De novo protein synthesis analysis by SUnSET assay in LN18 cells treated with control (siCtr), hnRNP H and/or hnRNP F (siH, siF or siH/F), hnRNP A1 (siA1) and hnRNP I (siI) siRNAs for 72 h, followed by Western blot analysis of the incorporated puromycin, hnRNP H/F, hnRNP A1, hnRNP I and GAPDH. Shown is a representative result from *n* = 4 for siH, siF and siH/F or *n* = 3 for siA1- and siI-independent experiments. (**D**) Quantification of puromycin incorporation from (**C**). For panels (**B**,**D**): blue, green and purple bars are respectively linked to hnRNP H, hnRNP F and hnRNP H/F; data are presented as mean values ± SEM of *n* = 3 independent experiments; * *p* < 0.05; ** *p* < 0.005; *** *p* < 0.0005; NS—non-significant (two-sided paired *t*-test).

**Figure 4 cancers-14-01283-f004:**
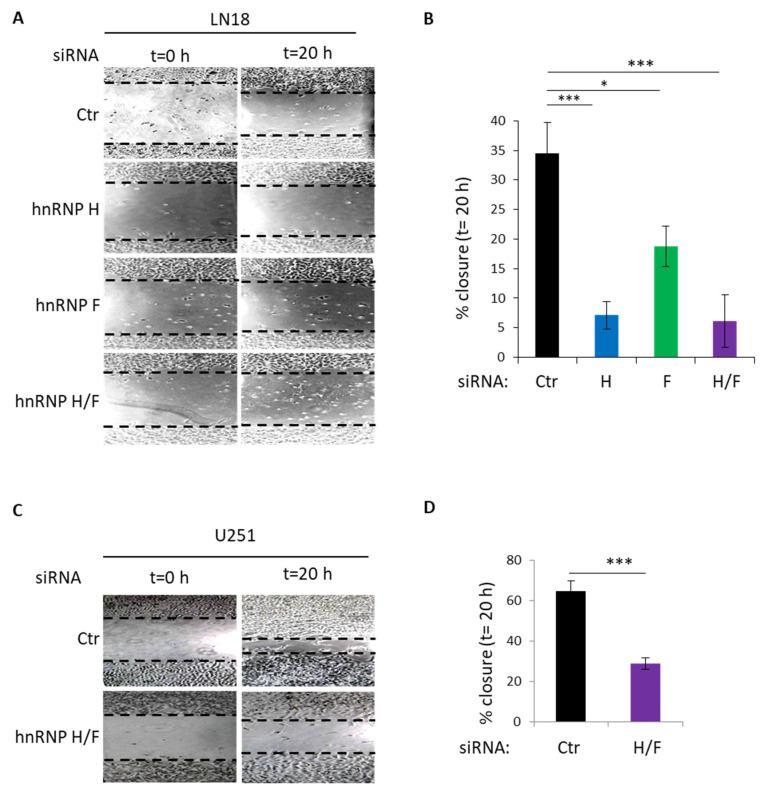
hnRNP H/F depletion inhibit GBM cancer cell migration. (**A**,**C**) Representative images of scratch wound assay, immediately (t = 0 h) or 20 h (t = 20 h) after the scratch, performed on monolayer of (**A**) LN18 or (**C**) U251 cells treated with 2.5 nM of siRNA control (siCtr), siRNAs against hnRNP H and/or hnRNP F (siH, siF and siH/F) for 72 h. (**B**,**D**) Percentage of wound closure measured 20 h (t = 20 h) after wound, at consistent locations. Data are presented as mean values ± SEM of *n* = 3 independent experiments, * *p* < 0.05, *** *p* < 0.0005 (two-sided paired *t*-test). Blue, green and purple bars are respectively linked to hnRNP H, hnRNP F and hnRNP H/F.

**Figure 5 cancers-14-01283-f005:**
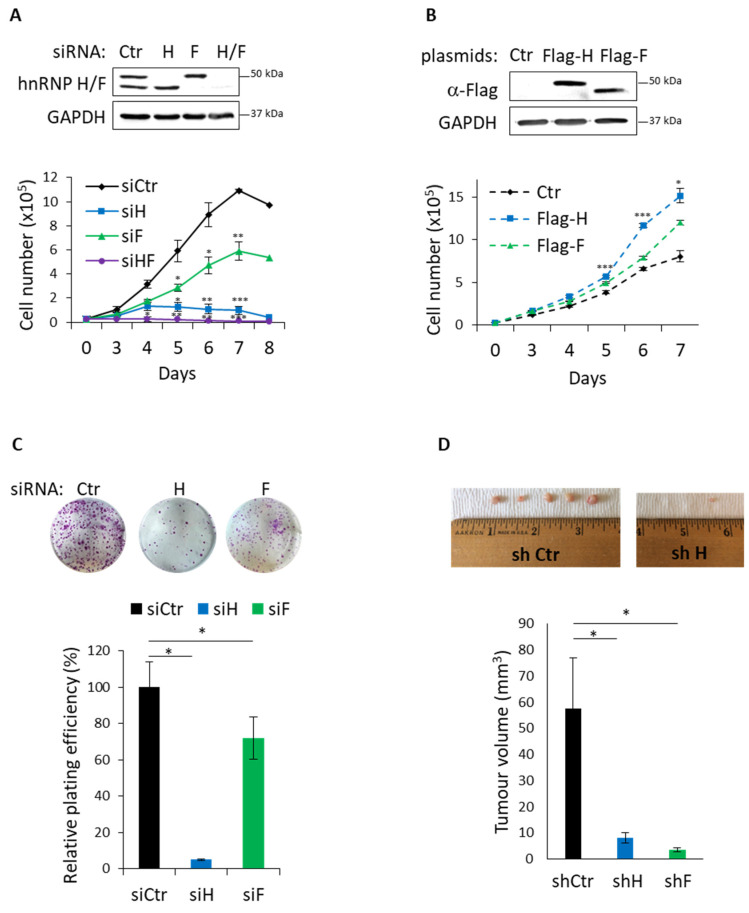
hnRNP H/F regulate GBM cancer cell proliferation in vitro and in vivo. (**A**,**B**) Proliferation assay and Western blot analysis of (**A**) LN18 cells treated with 2.5 nM of siRNA control (siCtr), siRNAs against hnRNP H and/or hnRNP F (siH, siF or siH/F) or (**B**) stable U87 cells expressing a control plasmid (Ctr), Flag-hnRNP H (Flag-H) or Flag-hnRNP F (Flag-F). Data are presented as mean values ± SEM of *n* = 3 independent experiments. (**C**) Representative images and quantification of colony foci formation in a monolayer culture of LN18 cells treated with 2.5 nM of siRNA control (siCtr), siRNAs against hnRNP H and/or hnRNP F (siH, siF or siH/F). Data are presented as mean values ± SEM of 5 independent wells. (**D**) Images and quantification of the volumes of the xenograft tumors formed in nude mice 40 days after the subcutaneous injection of stable LN18 cells expressing shRNA control (shCtr), shRNA against hnRNP H and hnRNP F (shH and shF). Data are presented as mean values ± SEM of *n* = 4 mice per group with bilateral flanked tumors per mouse, * *p* < 0.05 (two-sided paired *t*-test). For all the panels, * *p* < 0.05, ** *p* < 0.005 and *** *p* < 0.0005 (two-sided paired *t*-test).

**Figure 6 cancers-14-01283-f006:**
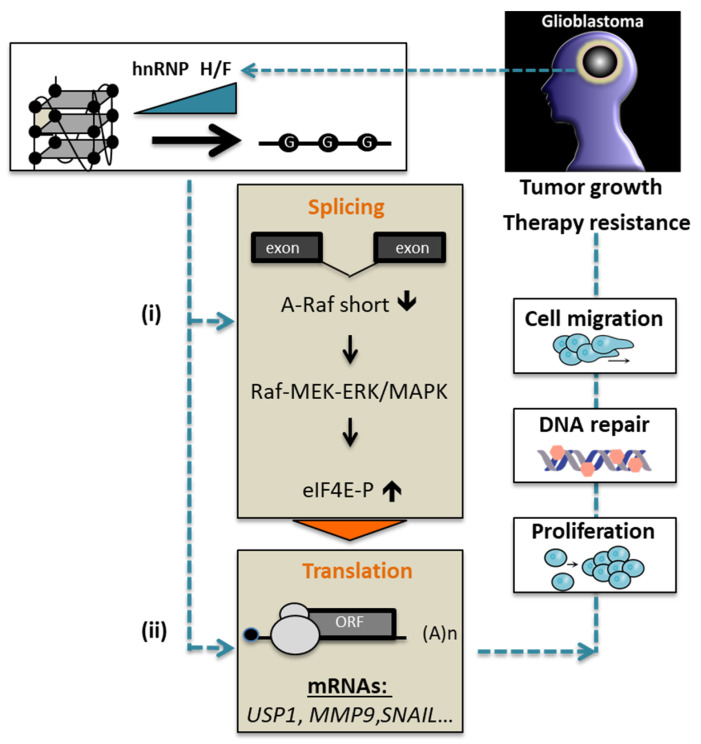
Model for the role of hnRNP H/F in regulating mRNA translation in GBM. Increased expression of hnRNP H/F in GBM [26] indirectly (i) and/or directly (ii) regulates mRNA translation through different overlapping mechanisms, including: (i) *A-Raf* mRNA splicing that in turn modulates the MEK-ERK/MAPK pathway to control the phosphorylation of eIF4E (this work) known to impact on translation, and (ii) the direct involvement of hnRNP H/F in translation of specific mRNAs [26]. The underlying regulatory mechanisms involve modulation of RG4 folding by hnRNP H/F and result in both global and selective control of translation impacting key functional cellular pathways, such as cell migration, proliferation and DNA repair, involved in GBM tumor growth and resistance to treatments.

## Data Availability

Data are available upon reasonable request.

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
