# Peer review of "Translational Regulation by hnRNP H/F Is Essential for the Proliferation and Survival of Glioblastoma"

_cancers, 2022, doi:10.3390/cancers14051283_

Round 1

Reviewer 1 Report

Although the direct link between A-Raf splicing and the phosphorylation of ERK and eIF4E is missing, I accept the authors' justifications and have no further comments.

Reviewer 2 Report

Following the revision work, the manuscript has been improved and at this step I recommend it for publication

This manuscript is a resubmission of an earlier submission. The following is a list of the peer review reports and author responses from that submission.

Round 1

Reviewer 1 Report

This is an excellent study into the role of RNA binding proteins on GBM tumorigenesis with emphasis on the hnRNP H/F - A-Raf pathway.

The only issue I have with the methodology used is the xenograft subcutaneous implantation - was consideration made for use of an orthotropic model?

Otherwise I feel this manuscript makes a significant contribution to our understanding of the above processes and the potential clinical impact of therapies targeting translational regulatory pathways in GBM patients.

Reviewer 2 Report

overall comments: The author found that hnRNP H/F regulate the RNA splicing of A-Raf kinase, the phosphorylation of eIF4E through RG4-dependent mechanism, hnRNP H/F are essential for key functional pathways regulating proliferation and survival of GBM. However, there is no direct evidence to show A-Raf is a regulator to control eIF4E phosphorylation. I am not very convinced that there is a tight link between these targets and the proliferation and survival of GBM. The quality of some data is hard to evaluate.

Point 1: Figure 1A: There are three isoforms of HnRNP H, which one is the most important one to regulate the A-Raf splicing in this work.

Point 2: Figure 1A: Please provide the protein level of both short and long isoforms.

Point 3: Figure 1B: Most of HnRNP H/F are localized in the nucleus, and the RNA processing including splicing also occurred in nucleus. Why the author utilizes the cytoplasmic extract to perform the HnRNP H/F IP enrichment assay to check the HnRNP H/F binding across A-Raf mRNA.

Point 4: Figure 1B: Please also provide the western blot results of nu/ce extraction as well as the HnRNP H/F IP enrichment.

Point 5: Figure 1C: A-Raf WB should be provided

Point 6: Line 94: There is no direct evidence to show A-Raf is key to control eIF4E phosphorylation. I would suggest checking the eIF4E phosphorylation with overexpression of A-Raf short or longer isoform or specific depletion one of the isoforms.

Point 7: Figure 2B: A-Raf WB should be provided.

Point 8: Figure 3D: The de novo protein synthesis performed by depleted HnRNPA1 is better than lane H in Figure 3C. The difference is very small between lane ctr and lane H. Please explain why there is no significance in lane A1, but with a strong significance in lane H.

Point 9: Figure S1B: How about the RNA expression of the longer isoform?

Point 10: Figure S2A: From the WB result, depletion H would increase the level of F, vice versa. There is no difference in the main figures, please clarify.

Point 11: Figure S2A: The depletion efficiency with H#2 is only a little or no. Please add strong evidence to avoid the siRNA off target.

Point 12: Lane 490-491: “4E-BP phosphorylation (Figure S3)”, missed the wb of 4E-BP phosphorylation

Point 13: Figure S6A: The panel is so small, hard to read.

Reviewer 3 Report

The work of Le Bras et al. is a thoroguh functional characterization of the role of hnRNP H/F, RNA-binding proteins acting as regulators of translation, in regulating proliferation, aggressiveness and survival of GBM. The work builds on previous observations done by the same authors and published last year in Nature Communications, that hnRNP H/F are binders of the unfolded state of RNA G-quadruplexes (RG4) structures of mRNAs of cancer-related genes. The authors here uncover an interesting axis whereby hnRNP H/F control the splicing of A-raf via RG4 binding, which in turn regulates the phosphorylation of eIF4E and thus the translation of all proteins regulated by eIF4E. More generally, the authors find by using SUNSET assay and polysome profiling, that hnRNP H/F influence the overall process of translation of glioblastoma cells via additional, yet to be identified mechanisms. The experiments provided are convincing as is the overall story. Some minor concerns:

1) a schematic illustration of the working hypothesis (based on previous publication) and the results obtained in this work, could really help the understanding of the message to readers that are not necessarily experts in RBPs, but are interested in the discovery of new targets to fight GBM.

2)  Fig. 5D: the representative figure of shF tumor xenograft is missing, while the quntification is provided in the graph below.

3) Fig. S5B: check the Y axis title; furthermore: why is the variation of siHF treatment not significant versus the siCtr one, while in the text it is described as the sum of siH and siF treatments? 

4) the authors should carefully check for english grammar errors and typos, e.g. second line of the introduction tumor -> tumors; first line of paragraph 3.2: remove comma after Since; the last sentence of the simple summary should be revisited. 

Round 2

Reviewer 2 Report

The author answered most of the concerns, but still did not make a tight link between the splicing and the consequence in GBM. Here I pointed the unanswered part which is important and reasonable.

Point2: As the author mentioned that there is no commercially antibody against short A-Raf, but there still have been detected with the antibody in some of the group. The study focused on the splicing and its consequences in GBM, we already see the change of the splicing in the RNA level, but the function of the gene should be carried out by the protein. Without the protein result, we cannot know how and to what extent of the protein level (short and longer) will be affected with the depletion of HnRNP H/F and the corresponding consequences to the GBM cancer cell migration and proliferation.

Point3: The study focused on the splicing of A-Raf by HnRN H/F, which may have other function, but definitely not the splicing in cytoplasmic. The concern is that although we see the enrichment of binding on both isoforms in cytoplasmic extracts, but what is function of the binding of HnRNP /F on the mature isoform. The splicing of the A-Raf is already finished in nucleus.

Point6: The author utilizes the HnRNP H/F siRNA or RG4 ligands to mimic the effect of HNRNP H/F depletion on G-quadruplex formation and splicing to get the similar results on ERK and 4E phosphorylation. The results will be obtained by the consequence of splicing change, A-Raf may be one, not the only one of the splicing targets of HnRNP H/F. Without the overexpression of A-Raf short or longer isoform or specific depletion one of the isoforms, how do we know the results on ERK and 4E phosphorylation are from the splicing of A-Raf specifically? How do the author answer the result “3.1. hnRNP H/F impact on eIF4E phosporylation via A-Raf splicing”?

Point 11: Please provided the normalized results of the WB, since the loading control in the H#2 lane is dramatically less than other lane and the quantification of puromycin incorporation

Author Response

Thank you for the opportunity to submit a revised version of our manuscript “Translational regulation by hnRNP H/F is essential for proliferation and survival of glioblastoma”.

Please find our point by point response in the word document joined.
